# Induction of a chromatin boundary in vivo upon insertion of a TAD border

Andréa Willemin[1‡¤a], Lucille Lopez-Delisle[2‡], Christopher Chase Bolt[2], Marie-Laure Gadolini[1], Denis Duboule[1,2,3‡]*, Eddie Rodriguez-Carballo[1‡¤b]*

1 Department of Genetics and Evolution, Faculty of Science, University of Geneva, Geneva, Switzerland, 2 School of Life Sciences, Ecole Polytechnique Fédérale de Lausanne, Lausanne, Switzerland, 3 Collège de France, Paris, France

¤a Current address: Epigenetic Regulation and Chromatin Architecture Group, Berlin Institute for Medical Systems Biology, Max Delbrück Center for Molecular Medicine (MDC), Berlin, Germany.
¤b Current address: Department of Molecular Biology, Faculty of Science, University of Geneva, Geneva, Switzerland.
‡ AW and LL-D are co-first authors on this work. DD and ER-C are co-last authors on this work.
* Denis.Duboule@unige.ch (DD); Edgardo.Rodriguez@unige.ch (ER-C)

**Data Availability Statement:** All datasets produced in this study were deposited in the Gene Expression Omnibus (GEO) under the accession number GSE166584.

## Abstract

Mammalian genomes are partitioned into sub-megabase to megabase-sized units of preferential interactions called topologically associating domains or TADs, which are likely important for the proper implementation of gene regulatory processes. These domains provide structural scaffolds for distant *cis* regulatory elements to interact with their target genes within the three-dimensional nuclear space and architectural proteins such as CTCF as well as the cohesin complex participate in the formation of the boundaries between them. However, the importance of the genomic context in providing a given DNA sequence the capacity to act as a boundary element remains to be fully investigated. To address this question, we randomly relocated a topological boundary functionally associated with the mouse *HoxD* gene cluster and show that it can indeed act similarly outside its initial genomic context. In particular, the relocated DNA segment recruited the required architectural proteins and induced a significant depletion of contacts between genomic regions located across the integration site. The host chromatin landscape was re-organized, with the splitting of the TAD wherein the boundary had integrated. These results provide evidence that topological boundaries can function independently of their site of origin, under physiological conditions during mouse development.

## Author summary

During development, enhancer sequences tightly regulate the spatio-temporal expression of target genes often located hundreds of kilobases away. This complex process is made possible by the folding of chromatin into domains, which are separated from one another by specific genomic regions referred to as boundaries. In order to understand whether such boundary sequences require their particular genomic contexts to achieve their isolating effect, we analyzed the impact of introducing one such boundary, taken from the

**Funding:** DD received support under Grant No. 310030B_138662 from the Swiss National Research Foundation (http://www.snf.ch), Grant SystemHox No. 232790 from the European Research Council (https://erc.europa.eu), Grant RegulHox No. 588029 from the European Research Council (https://erc.europa.eu). L. L-D. was supported by the ERC grant RegulHox (No 588029, to D.D.) C. C. B. by a fellowship of the Eunice Kennedy Shriver National Institute of Child Health & Human Development of the National Institutes of Health (No F32HD093555, to C.C.B.) https://www.nichd.nih.gov. The funders had no role in study design, data collection and analysis, decision to publish, or preparation of the manuscript.

**Competing interests:** The authors have declared no competing interests exist.

*HoxD* locus, into a distinct topological domain. We show that this ectopic boundary splits the host domain into two sub-domains and affects the expression levels of a neighboring gene. We conclude that this sequence can work independently from its genomic context and thus carries all the information necessary to act as a boundary element.

## Introduction

Inside the cell nucleus, mammalian genomes are organized at various levels or resolution, from the nucleosomal scale to chromosome territories [1]. At the intermediate level, the use of whole-genome chromosome conformation capture techniques (such as Hi-C) in interphase cells identified sub-megabase to megabase (Mb) structures referred to as topologically associating domains (TADs). These domains appear as discrete on-diagonal pyramid shapes in Hi-C maps, reflecting a high frequency of internal interactions, which seemingly participate in enhancer-promoter communication [2,3]. The limits between TADs are usually referred to as boundaries (or borders) and display variable strengths in terms of contact blockage, often expressed as their insulation score [4]. In the vast majority of cases, TAD boundaries host binding sites for CTCF and they are associated with other features including housekeeping genes and CpG islands [2,5,6].

In vertebrates, the CTCF protein and the cohesin complex participate in DNA interactions, likely through a loop extrusion mechanism: once loaded onto the DNA, the cohesin complex extrudes chromatin, a process stabilized or stalled whenever the cohesin complex encounters bound CTCF sites, preferentially of convergent orientation, or when two loops collide [7,8]. Although the depletion of these proteins or some of their co-factors alters the formation of loops and TADs genome-wide, it seems to have only minor effects on gene expression [9–14], raising questions regarding the impact of chromatin structure upon genome function [15]. In this context, it was proposed that chromosome topology may refine the action and timing of distant enhancers on their target genes during development [16–18], implying that the importance of such structures should be considered on a case-by-case basis, rather than drawing too global conclusions.

A useful experimental approach to study TADs and their boundaries in a locus-specific manner is to engineer alleles with deletions of specific elements. In some loci, deletions of boundaries or rearrangements of TADs were associated either to cancer [19] or to developmental defects, as seen for example with *Xist* [3], *Wnt6/Ihh/Epha4/Pax3* [20], *HoxD* [21,22], *Firre* [23], *Sox9/Kcnj2* [24] or *Shh* [25]. In contrast, while fewer examples exist where TAD boundaries were inserted into specific genomic locations, they showed different levels of impact upon the surrounding chromatin environment [23,26,27]. Therefore, the capacity of some DNA sequences to act as TAD borders and their effect on gene regulation might in part depend on the host genomic context. Alternatively, the ability of a given TAD boundary element to delimit a chromatin domain may be mostly encoded in its underlying sequence.

The TAD structure of the *HoxD* locus has been studied in some details. The *HoxD* gene cluster itself contains a strong TAD boundary separating two distinct regulatory landscapes referred to as C-DOM and T-DOM, each hosting series of enhancers. The T-DOM TAD is further divided into two sub-TADs at the level of a DNA segment called CS38-40, which contains a limb enhancer, a CpG island in close proximity to the transcription start site (TSS) of the *Hog* and *Tog* long non-coding RNAs (lncRNAs) and three occupied CTCF sites all oriented towards the gene cluster [21,22,28]. This region helps to properly implement the timing of limb enhancer action and constitutes a bona fide topological boundary, for its deletion

abrogates the observed contact segregation, whereas its inversion reinforces it [18]. We wondered whether this boundary region would by itself carry the capacity to create a topological boundary when positioned into a different genomic context *in vivo*, and hence we generated transgenic mouse models with random ectopic integrations of this region. We report the ability of this region to be accessed by both CTCF and the cohesin complex, and show that this boundary was able to split the 1.2 Mb-large host TAD into two sub-structures. These topological changes were accompanied by a decrease in the expression of the *Btg1* gene, the only protein-coding gene present inside the host TAD, further illustrating the functional impact of introducing CS38-40 at an ectopic site.

## Results

### Region CS38-40 is a sub-TAD boundary of the *HoxD* locus

Through series of deletions and inversions *in vivo*, we recently showed that region CS38-40, located within the large T-DOM TAD flanking the *HoxD* cluster, was capable to act as a topological border [18]. We further evaluated the properties of this region by generating Hi-C datasets using mouse limb cells at embryonic day 12.5 (E12.5) in a mutant line bearing a deletion of CS38-40 (*del(CS38-40)*) [18,29] along with wild-type control cells. We identified topological domains (TADs or sub-TADs) using the insulation-based hicFindTADs algorithm and confirmed the partitioning of the T-DOM TAD into two sub-domains at the level of CS38-40 in wild-type limbs. This boundary however appeared relatively weak and was only identified with the smallest window size that was applied (240 kb) (Fig 1A and 1C, light blue bars and track; S1 Table). Using the same algorithm and parameter values, we did not detect the splitting of the T-DOM in *del(CS38-40)* (Fig 1B, light red bar and S1 Table), demonstrating the merging of the two T-DOM sub-TADs into one single structure. Taken together, these results confirm that region CS38-40 is a bona fide sub-TAD border, validating previous observations [18,21,28].

Region CS38-40 contains three highly conserved non-coding sequences (CS38, CS39 and CS40) as well as a limb enhancer (comprised within CS39) and three CTCF sites oriented towards the *HoxD* cluster (Fig 1D and 1E). To evaluate the capacity of region CS38-40 to function as a sub-TAD boundary outside of its original genomic context, we performed random transgenesis by pronuclear injection of a 45 kb-large fosmid clone containing the entire region CS38-40 (Figs 1E and 2A). Because transgenes often integrate in multiple copies [30], we used a *loxP*/Cre system [31] to try and reduce the copy number to one and obtained a stable transgenic mouse line termed *TgN(38–40)*.

### Characterization of the TgN(38–40) integration

To locate the TgN(38–40) integration site, we carried out targeted locus amplification (TLA; Fig 2B) [32]. TLA is a 3C-derived technique based on fixation, enzymatic digestion, proximity ligation and inverse PCR from a specific locus of interest (viewpoint) that results in an overrepresentation of the sequences surrounding the viewpoint. Given the large ligation products of TLA (~2 kb), it is suitable for characterizing transgene-integration sites and chromosomal rearrangements with a base-pair resolution [32–34]. We used TLA viewpoints corresponding to two regions of the transgene, one located in the vector backbone and the other matching the CS38 element. A primary TLA analysis assigned the integration site to a region located in chromosome 10 (Fig 2C, top two tracks and S1A Fig) and identified one of the integration breakpoints based on abrupt drops in the coverage of reads mapped in local mode (similar to [35], but with more filters; see the Materials and Methods section and examples in Fig 2C). This breakpoint connected the CS38 element to chromosome 10 (S3B and S3C Fig, TLA-

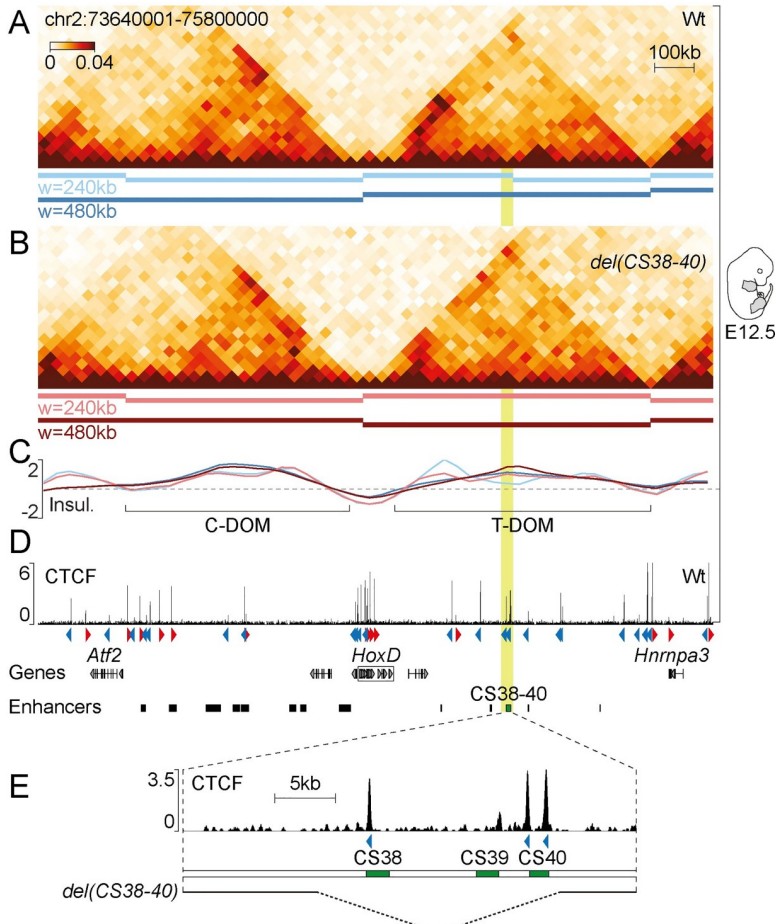

**Fig 1. Region CS38-40 is a sub-TAD boundary of the *HoxD* locus.** (A, B) Hi-C of the *HoxD* locus in wild-type (A) and *del(CS38-40)* (B) whole limbs at E12.5. Topological domains (TADs or sub-TADs) were identified using a window size (w) of 240 kb (light-coloured bars) or 480 kb (dark-coloured bars) and are represented below each Hi-C heatmap. (C) Insulation scores computed with two different window sizes (see above). (D) Wild-type ChIP-seq of CTCF in E12.5 whole limbs. CTCF orientations are indicated by red or blue arrowheads. (E) Magnification of region CS38-40 (mm10, chr2:75122684–75160161; highlighted in light green in previous panels) from panel D. The extension of the deletion in *del(CS38-40)* (mm10, chr2:75133816–75153815) is shown as a dashed line at the bottom.

identified right breakpoint). In addition, mapping the TLA data on the transgene sequence showed coverage along the whole construct (S1B Fig) and multiple reads supported a tail-to-head tandem junction, whereas none supported tail-to-tail nor head-to-head configurations (S1C Fig). Therefore, these results strongly suggested the integration of at least one full copy of the fosmid, fused tail-to-head with a truncated version of a second copy.

We then tried to determine how many copies of the TgN(38–40) construct were left after the *Hprt^cre* cross and performed transgene quantification by qPCR on purified genomic DNA (gDNA) (Figs 2B and S2A). We compared samples that were hemizygous for the integration (*TgN(38–40)/Wt*) to wild-type (*Wt*) controls. To validate the qPCR approach, we used samples heterozygous for the deletion of the endogenous region CS38-40 in chromosome 2 (*del(CS38-40)^+/-^*) and a qPCR target region located outside CS38-40 (*Hoxd8d9*), which should not show amplification differences in any sample. In *TgN(38–40)* hemizygous animals, for which two copies of CS38-40 were attributed to the endogenous locus (reflecting the two non-deleted alleles), the ectopic (surplus) values of CS38, CS39 and CS40 were 1.46, 0.55 and 0.81,

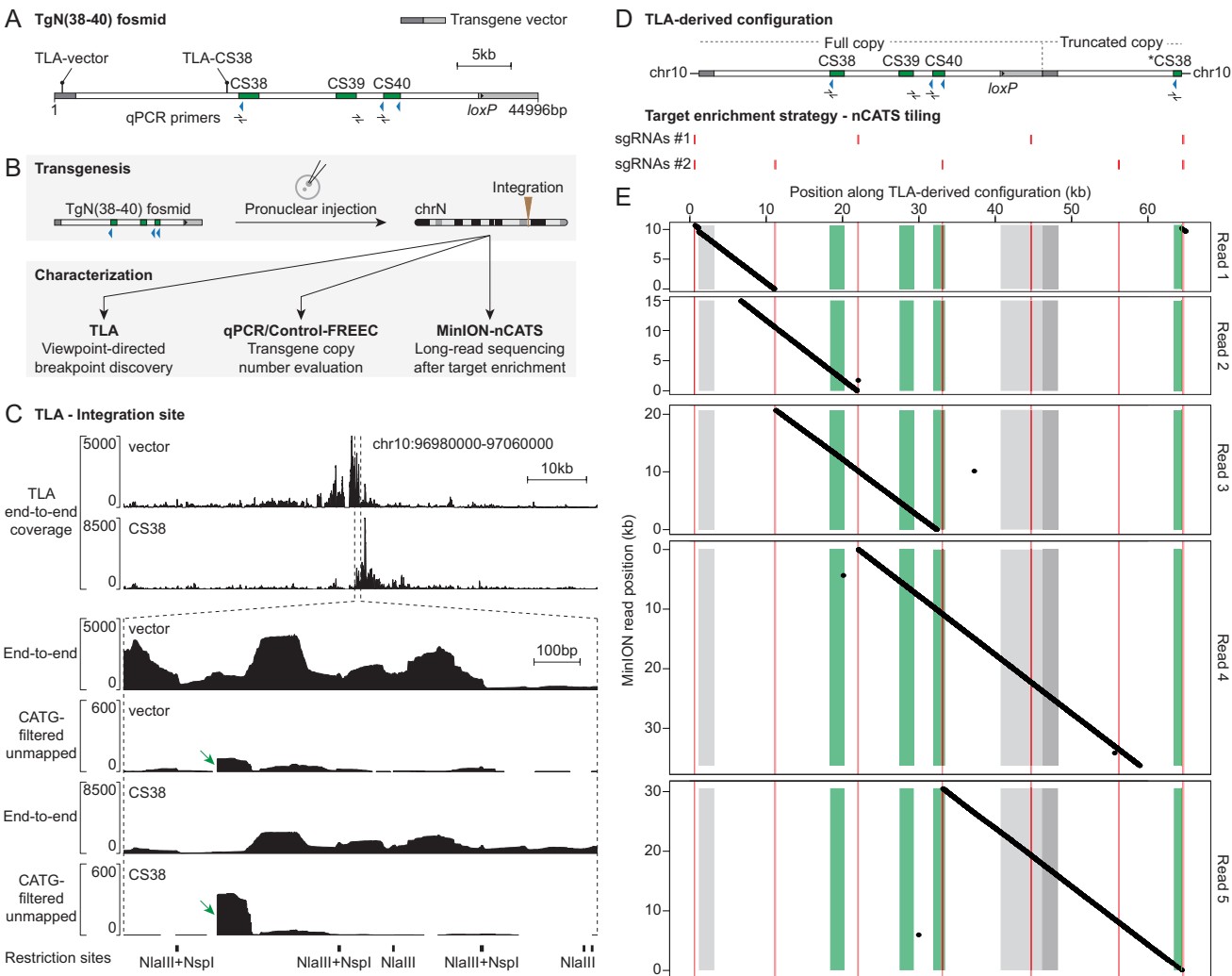

**Fig 2. Characterization of the TgN(38–40) integration.** (A) Schematic representation of the TgN(38–40) fosmid, vector in gray. Position of TLA viewpoints (lollipops) and qPCR primers (left to right: CS38, CS39 and CS40a) is indicated. (B) Transgenesis and characterization workflow. (C) Analysis of the TgN(38–40) integration by TLA. Top, TLA end-to-end coverage around the candidate integration site. Bottom, magnification of the above-mentioned area. Both end-to-end and CATG-filtered unmapped coverages are shown. Green arrows highlight sharp drops in the CATG-filtered unmapped signal. TLA restriction sites are shown below. (D) Top, TLA-derived genomic configuration. Asterisk, second partial CS38 element. Bottom, position of the two distinct pools of sgRNAs used for nCATS. (E) Analysis of the TgN(38–40) integration using MinION-nCATS. Dot plot representation of five selected MinION reads (ranging from 10 to 35 kb) along the TLA-derived configuration (horizontal axis), color-coded as in (D). The ~600 bp duplication of chromosome 10 sequence manifests as a subtle discontinuity in read 1 on the left as well as a small diagonal on the right.

respectively (S2A Fig), thus showing that these elements were represented in variable copy numbers in the transgene.

We also used the control-free copy number and allelic content caller (Control-FREEC) (Figs 2B and S2B) [36], a maximum likelihood-based algorithm that evaluates copy number along genomic regions starting from NGS data. We applied Control-FREEC to gDNA libraries from specimens that were both hemizygous for the TgN(38–40) integration and homozygous for the endogenous CS38-40 deletion (*TgN(38–40)/Wt; del(CS38-40)$^{-/-}$*; referred to as 'test') as well as from control samples to represent the endogenous region CS38-40 in chromosome 2. We obtained a copy number estimation close to four (using two different windowing functions) for the segment extending from the left end of the TgN(38–40) construct towards the

CTCF site of CS38 (S2B Fig). Taken together, both the qPCR and Control-FREEC results indicated that the insert consisted of a single complete copy of region CS38-40, followed by a fragment extending towards a second but partial CS38 element. We thus constructed an *in silico* mutant genome composed of one entire copy of region CS38-40 and a partial copy including a truncated CS38 element in chromosome 10 (Fig 2D).

To validate this conclusion and to characterize the missing integration breakpoint, we implemented nanopore Cas9-targeted sequencing (MinION-nCATS; Fig 2B, 2D and 2E) [37]. We targeted two distinct combinations of single-guide RNAs (sgRNAs) around the insertion site and along the transgene (at approximately 10 kb intervals) in order to release overlapping DNA fragments ranging from 9.5 to 23 kb in size (Fig 2D; see nCATS tiling). The MinION sequencing reads were mapped onto the above-mentioned *in silico* mutant genomic configuration (Fig 2D).

The MinION coverage revealed an around 25 times enrichment of sequences originating from the targeted region compared to the rest of the genome (S4 Table). Inspection of five individual MinION reads enabled us to map the entire transgene integration (Fig 2E) and confirmed the presence of the additional partial copy of the construct that includes a second CS38 segment (with its CTCF site; see Fig 2E, read 5). Moreover, MinION unveiled a ca. 600 bp duplication of chromosomal sequence and primarily identified the missing (left) integration breakpoint in between the duplicated segments (Fig 2E, read 1). These results prompted us to design a TLA breakpoint analysis pipeline that considered more reads, which enabled the base-pair mapping of the left breakpoint (S3A Fig, red arrows; and S3B and S3C Fig, MinION-identified left breakpoint). Therefore, we established that the insertion of the TgN(38–40) construct in chromosome 10 resulted in a partial tail-to-head tandem, which consisted of one entire copy of the construct fused with an additional fragment including the CTCF site of the partial CS38 element. As a consequence, the insert spans 63.2 kb in total and comprises four CTCF sites, which are all sharing the same orientation. Both the TLA and MinION results indicated that the integration of the construct was not associated with any major genetic reshuffling of the host locus, except for the small duplication described above (S3C Fig).

## Recruitment of architectural proteins on the relocated region CS38-40

We looked for the presence of both CTCF and the cohesin subunit RAD21 on the ectopic region CS38-40 by chromatin immunoprecipitation (ChIP) coupled with sequencing using whole limbs of *TgN(38–40)* embryos at E12.5 (Fig 3A). The transgene was brought on top of a deletion of both CS38-40 endogenous copies, such that all potential sequencing reads would derive from the transgenic locus. Mapping of the ChIP data onto the mm10 reference genome revealed the binding of CTCF on all three sites of CS38-40, as in the wild-type situation (Fig 3A, CTCF). The signal at the CTCF site of CS38 approximately corresponded to twice the signal at either of the other two sites, probably due to the additional copy (Fig 3A, see control regions in S4 Fig). Similar to what was observed for the control endogenous CS38-40 region, RAD21 was mostly enriched on the CS38 site (Fig 3A, RAD21). These results indicated that the recruitment of architectural proteins on region CS38-40 could occur independently from the global genomic context.

## Alteration of local chromatin structure upon integration of the TgN(38–40) construct

We next investigated the conformational state of the region hosting the TgN(38–40) construct in chromosome 10 using our control Hi-C dataset and observed the presence of a 1.2 Mb-large, well defined TAD wherein the transgene had integrated (Fig 3B). This domain contains

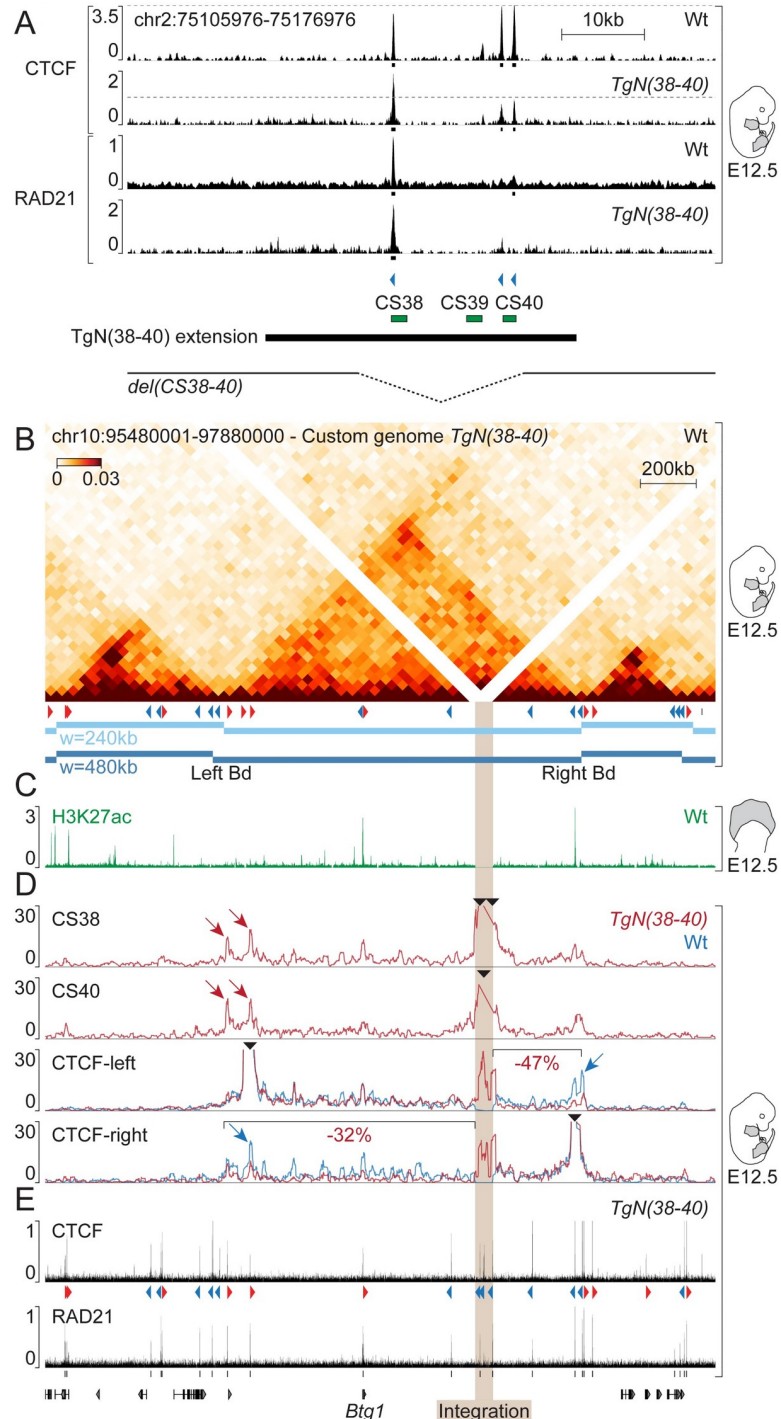

**Fig 3. Recruitment of architectural proteins and topological changes upon integration of the TgN(38–40) construct.** (A) ChIP of CTCF and RAD21 in wild-type or *TgN(38–40)* E12.5 whole limbs. The window displayed corresponds to the native region CS38-40. Dashed lines are displayed for better comparison between the occupancy of various CTCF sites. Peak calling is represented as black boxes. Bottom, extension of both the TgN(38–40) construct and *del(CS38-40)* background. (B) Hi-C showing the host locus of chromosome 10 in wild-type whole limbs at E12.5. Below the Hi-C heatmap, wild-type CTCFs (red or blue arrowheads) and topological domains (horizontal bars). (C) Distribution of H3K27ac (green) over the host locus in the distal part of wild-type E12.5 forelimbs. (D) Top, 4C-seq using CS38 and CS40 as viewpoints in *TgN(38–40)* E12.5 limbs. Bottom, 4C-seq tracks of CTCF-left and CTCF-right viewpoints in both *TgN(38–40)* homozygous (red lines) and wild-type samples (blue lines). Percentages of 4C-seq

contact changes beyond the integration site are shown. Black arrowheads indicate 4C-seq viewpoints. (E) ChIP of CTCF and RAD21 over the host landscape in *TgN(38–40)*.

relatively few CTCF sites (~5 sites/Mb) as well as a single gene, *Btg1* (Fig 3, bottom). Examination of published ChIP-seq datasets for the active histone mark H3K27 acetylation (H3K27ac) in E12.5 distal limb cells [22] revealed two strongly acetylated regions within the TAD, corresponding to either the promoter of the *Btg1* gene, or a region at the telomeric (right) TAD border (Fig 3C).

We then assessed whether and how the four ectopic CTCF-binding sites would interfere with the host chromatin landscape, knowing that all four had the same orientation. We performed circular chromosome conformation capture-sequencing (4C-seq) using the CTCF sites present in the CS38 and CS40 elements as viewpoints in *TgN(38–40)* transgenic samples (Fig 3D, CS38 and CS40 tracks). Both CS38 and CS40 CTCF viewpoints established strong interactions with regions of their new genomic environment. Contacts were particularly frequent within the limits of the 1.2 Mb TAD hosting the transgene, suggesting that the surrounding landscape could constrain interactions originating from the ectopic DNA segment. Furthermore, maximum interaction frequencies were observed at CTCF sites displaying a convergent orientation relative to those of the transgene (Fig 3D, red arrows), in agreement with the loop extrusion model [7,8].

As additional viewpoints, we used two CTCF sites, also bound by RAD21 (Fig 3E), located at either extremity of the TAD (Fig 3D, CTCF-left and CTCF-right). In control limbs, both viewpoints established contacts mainly restricted to their own TAD (Fig 3D, blue tracks) and maximum contact frequencies were observed at convergent CTCF sites near the endogenous TAD boundaries (Fig 3D, blue arrows). To assess contacts in the *TgN(38–40)* samples without confounding effects due to the wild-type copy of this region, we carried out 4C-seq by using E12.5 limbs from embryos homozygous for *TgN(38–40)*. Novel contacts were observed between these two CTCF sites and the sites located within the TgN(38–40) transgene (Fig 3D, CTCF-left and CTCF-right, red tracks), corroborating the results obtained when both CS38 and CS40 CTCF sites were used as viewpoints. Of note, the new loops appeared to occur at the expense of endogenous interactions, since contacts established by each of the two endogenous viewpoints were decreased beyond the integration site relative to the position of the viewpoint. This was particularly evident at the boundaries of the TAD. This decrease in interactions relative to the control ranged from 32% to 47% for CTCF-right and CTCF-left viewpoints, respectively.

## Reconstitution of a sub-TAD boundary in the host landscape

These observations prompted us to evaluate whether the host TAD had been disrupted by the integration of the construct. We performed Hi-C in *TgN(38–40)* homozygous E12.5 limbs (Fig 4A–4C) and observed a split of the host TAD into two sub-domains, right at the level of the transgene insertion (Fig 4B, arrow). To better compare the wild-type and mutant Hi-C maps, we adapted the signal of the former to account for the potential effect of increasing the genomic distance. We thus calculated the value-of-alpha of the function relating contact frequencies to distance along the *Btg1* TAD (S5A Fig, pink curve). Direct comparison of wild-type and mutant datasets showed a clear loss of interactions (-39%, p-value = 2e-29) between the two new sub-domains in a differential heatmap (Fig 4D, dashed box). The TAD partitioning was reminiscent of a sub-TAD boundary formation, for it was only detected by the hicFindTADs algorithm at a window size of 240 kb (Fig 4B and 4C, light red bars and track, S1 Table) and a substantial amount of interactions were scored across the new border (Fig 4B; asterisk), similar

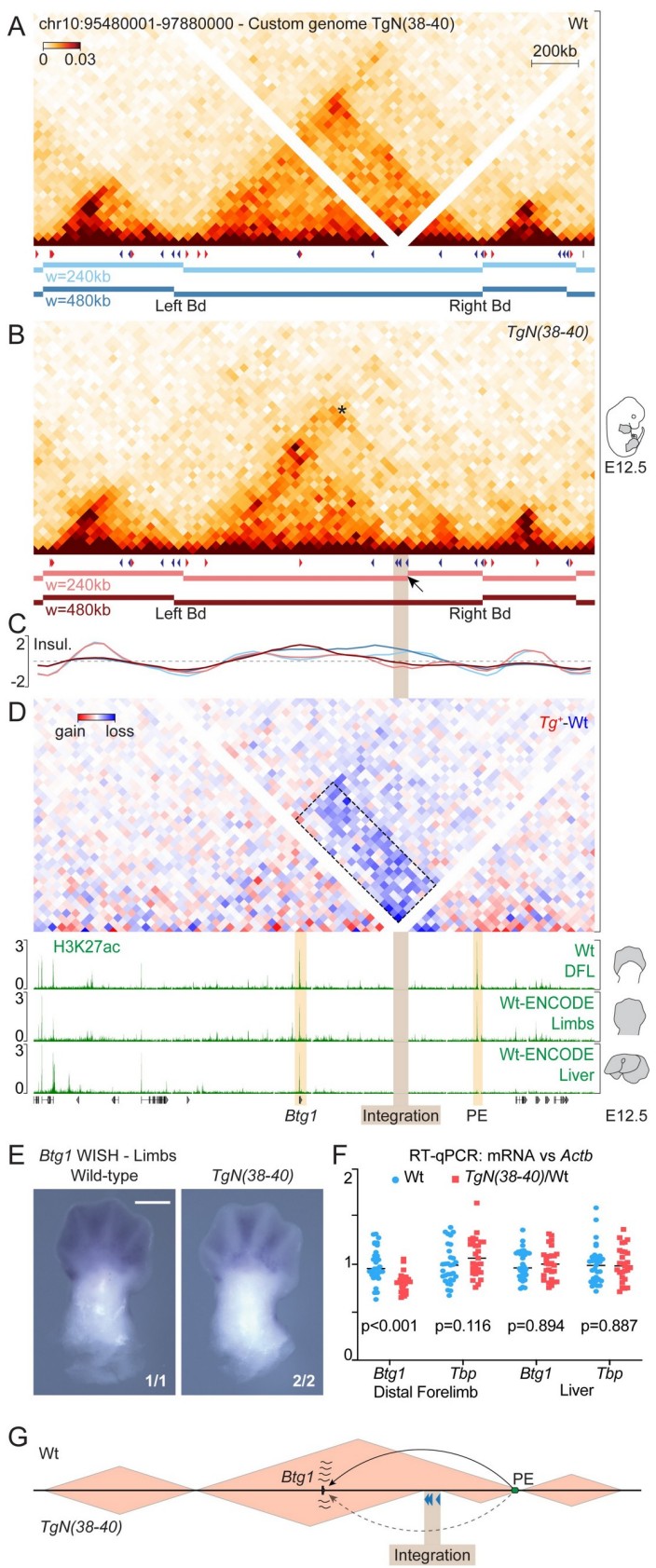

**Fig 4. Reconstitution of a sub-TAD boundary in the host landscape.** (A) Wild-type and (B) mutant *TgN(38–40)* Hi-C matrices of the host locus in whole limbs at E12.5. Corresponding CTCFs (red or blue arrowheads) and topological domains (horizontal bars) are shown. Below each panel, TAD-separation, color-coded according to the applied window size. The arrow indicates a new boundary at the level of the integration. (C) Insulation scores using two different window sizes. (D) Differential Hi-C heatmap (*TgN(38–40)-Wt*). Quantification of contacts changes between regions located across the TgN(38–40) integration site (dashed box), -39% (p-value = 2e-29). H3K27ac ChIP tracks. (E) *Btg1* WISH in wild-type and *TgN(38–40)* homozygous E12.5 forelimbs. Scale bar: 500 μm. Fractional numbers indicate the proportion of embryos displaying equivalent patterns in the experiment. (F) RT-qPCR values of *Btg1* and *Tbp* (internal control) in E12.5 distal forelimbs and liver. mRNAs levels were referenced to *Actb* (Wt = 31, *TgN(38–40)/ Wt* = 27); p-values were obtained by Welch's t-test. (G) Proposed mechanistic model of *Btg1* expression changes. PE, putative *Btg1* enhancer (green oval). Enhancer-promoter communication is represented with arrows.

to what is observed at the endogenous CS38-40 region (Fig 1A, S1 Table and [18,21,22]). Therefore, the integration of the TgN(38–40) construct not only impaired interactions between discrete loci that were separated by the insertion, but was indeed capable of reshaping the topological organization of the host landscape.

Finally, we determined whether the integration of the construct and associated chromatin reorganization would cause any modification in the expression of the *Btg1* gene (Figs 3 and 4), a gene involved in the regulation of cell proliferation [38]. Whole-mount *in situ* hybridization (WISH) of *Btg1* in E9.5 embryos showed an ubiquitous expression in wild-type and mutant embryos, with no significant increase of limb expression associated to the limb enhancer contained in the transgene (S6A Fig). At E12.5, control embryos also revealed a widespread expression, with maximum transcript levels in the developing limbs, facial mesenchyme, whisker pads, lateral plate mesoderm and mammary buds (Figs 4E and S6B and S6C). In *TgN(38–40)* homozygous embryos, *Btg1* appeared to be globally down-regulated, without any detectable morphological alteration (S6B Fig). The decrease in expression was particularly pronounced for the distal part of the limbs, the lateral plate mesoderm and the mammary buds (Figs 4E and S6B and S6C).

Further analyses by RT-qPCR confirmed that the insertion of the transgene had a negative effect on *Btg1* expression in distal limbs, yet not in liver cells (Fig 4F) where the isolated sub-domain did not show any H3K27ac signal at the putative regulatory region (Fig 4D, right orange bar). These results indicated that the integration of the TgN(38–40) construct and/or the associated reorganization of the host chromatin landscape had an impact on *Btg1* gene expression in a tissue specific-manner, most probably due to the isolation of a putative enhancer.

## Discussion

We report the ability of the region CS38-40 from the *HoxD* locus to function as a topological boundary when introduced outside of its original genomic context. To interpret the results in a reliable manner, we characterized the transgene integration at a base-pair resolution, assessing both the number of copies and the absence of major chromosomal rearrangements. We concluded that the fosmid clone was present in one copy, plus a truncated piece containing another CTCF site, leading to the presence of four CTCFs with the same orientation. Indeed transgenes tend to integrate as concatemers, up to hundreds of copies, with the most common configuration being in tail-to-head [39], a situation that would have invalidated the observed effects. Also, even though the correct insertion of large pieces of DNA (BAC or fosmids) as transgenes has been usually considered as granted, the results of such insertions have rarely been verified at the appropriate level of resolution. Our detailed characterization of the insertion site, by using various strategies [33–35,40], suggests that careful attention should be given to this aspect whenever using transgenic approaches to address questions related to chromatin organization.

## Reproducing a sub-TAD boundary

TAD boundaries have been deleted *in vivo* at several loci and these alterations were associated with changes in gene expression (reviewed in [41]), whereas in other instances, topological borders were moved along with part of their regulatory domains through targeted inversions [18,24]. Such inversions and repositioning of TAD boundaries led to a new spatial organization and induced the down-regulation of genes whose access to their enhancers was hampered. These situations, however, make it difficult to disentangle how much of the topological and functional effects are due to the specific positioning of the TAD border from the impact of the large rearrangement of the regulatory landscape; in other words, to which level the specific initial chromatin environment is by itself required for the function of the boundary.

To date, only few studies have undertaken the opposite approach, whereby a boundary is moved to a completely new genomic location, and most of them were limited to mammalian cultured cells. Furthermore, the results of these studies seem to be influenced by several factors including the nature of the sequence, the host environment, as well as more technical aspects. For instance, Redolfi et al. (2019) found that the introduction of a 2.7 kb piggyBac cassette containing three CTCF-binding sites led to the formation of new DNA loops and stripes [26]. Others reported a clear TAD splitting after insertion of the HERV-H transposon in an 8.7 kb piggyBac. This splitting, however, required the expression of the transposon, rather than high CTCF binding levels, which are associated with most canonical TAD boundaries [42]. In another transposon-based study, a 2 kb fragment containing a CTCF-binding site and a transcription start site (TSS) was inserted multiple times across the genome and several cellular clones were analyzed. The authors reported various degrees of topological changes including the generation of new loops and domains, compartmental changes and domain fusion, whereas in some instances, no change was observed. Of note, some of the effects resulted from the combined action of transcription and CTCF binding and could be modulated by the host genomic context [27].

The only *in embryo* attempt consisted in the integration of the cDNA from the *Firre* lncRNA, which harbored one CTCF-binding site and an inducible TSS. However, this construct was not able to induce TAD splitting in any of multiple integration sites assessed [23]. In our case, we previously showed that region CS38-40 is responsible for the organization of the *HoxD*-associated T-DOM in two sub-TADs and that the orientation of the CTCF-binding sites was important in this context [18]. We now report that this capacity is intrinsic to its underlying sequence, at least for the chromosomal context analyzed in chromosome 10.

It is noteworthy that the increase in the linear genomic distance induced by the integration (63.2 kb) may have an effect on the 3D chromatin organization in our experimental setup. Nevertheless, the evidence provided here suggests that the capacity of a given sequence to restrict contacts to one side does not strictly depend on the linear distance. Indeed, deletions of several DNA fragments of various lengths (and CTCF-binding sites) at the *HoxD* TAD boundary did not lead to a complete fusion of the two surrounding domains, but only increased the permeability of the boundary [22]. Similarly, the removal of the whole *Firre* locus, spanning 82 kb and comprising twelve CTCF-binding sites, left the two surrounding TADs almost unaffected [23].

## Disturbing a putative regulatory landscape

Our mouse model also allowed us to probe changes in gene expression in the vicinity of the integration site. The insertion of the TgN(38–40) construct took place in a large TAD that contains the single protein-coding gene *Btg1*, which has been implicated in maintaining the proliferation of neural stem cells [38]. We detected *Btg1* transcripts in several tissues, including the

facial region, the lateral plate mesoderm, the mammary buds and the limbs. A decrease of *Btg1* expression was evident upon TgN(38–40) insertion, particularly in the limbs. In limbs, apart from the *Btg1* gene itself, only a single other region appeared to be heavily decorated with H3K27ac, a histone mark associated with active genes and enhancers [43]. This H3K27ac-positive region was located at one of the TAD limits and became topologically isolated from *Btg1* upon integration of TgN(38–40). Conversely, in the developing liver, this region did not show remarkable H3K27ac signal and, concomitantly, *Btg1* mRNA levels were not affected by the integration of the transgene in this tissue. Although we cannot rule out other potential explanations, such as for example TgN(38–40) acting as a repressive regulatory sequence in this new context, we believe that the most plausible explanation for the downregulation of *Btg1* in limbs is its isolation from the sole putative limb enhancer found within the host TAD (Fig 4G).

## Materials and methods

A fully detailed version of all experimental procedures reported in this work can be found in [44] (https://zenodo.org/record/4292299).

### Ethics statement

All experiments of this study were accepted by the Geneva Cantonal committee for animal experimentation and were performed in accordance with the Swiss Animal Welfare Act (LPA) under the license no. GE 81/14 (to D.D.).

### Mutant mouse strains

The *TgN(38–40)* transgenic line was obtained by injecting the TgN(38–40) linearized fosmid (WI1-2299-I7; mm10, chr2: 75122702–75160145) into mouse fertilized oocytes at the pronuclear stage. 129S1/Sv-*Hprt^tm1(CAG-cre)Mnn^*/J (abbreviated *Hprt^cre^*) mice, described in [31], were purchased from The Jackson Laboratory and were used for the removal of extra-copies of the transgene in case of tandem integration thanks to a *loxP* site located in the transgene vector. The *HoxD^del(CS38-40)^* allele (abbreviated *del(CS38-40)*) was described previously [29]. All mutant mouse strains used in this study were maintained in a heterozygous state on a C57BL6xCBA background. Heterozygous individuals were crossed in order to generate embryos of all possible genotypes.

### TLA

TLA was performed as in [32] with the following adaptations. Limb cells were dissociated using collagenase type XI (Sigma-Aldrich, C7657) and the cell suspension was strained. Transgene-positive (*TgN(38–40)/-*) samples were identified by PCR and two E12.5 brains were used as starting material. TLA inverse PCR was performed using the viewpoint-directed inverse PCR primers listed in S2 Table. TLA library preparation was achieved using the Nextera DNA Flex Library Prep (Illumina) protocol, independently for each viewpoint. Libraries were sequenced as 100 bp single-end reads with an Illumina HiSeq 4000. TLA data analysis was performed using the custom pipelines described hereafter. In brief, all alignments were produced using Bowtie 2 (version 2.3.5) [45] and were sorted with SAMtools (version 1.9) [46]. After filtering and adapter trimming with cutadapt [47], the reads were mapped on their entire length (end-to-end) onto the mm10 reference genome or a sequence built with 4 copies of the fosmid with different orientations (->-><- ->) and the coverage was computed using BEDTools (version 2.27.1) [48]. The coverage from reads mapped on mm10 with a mapping quality (MAPQ) above 30 was assigned to non-overlapping 1 Mb windows of mm10 in order to

identify the candidate integration site as the genomic region with maximum TLA end-to-end coverage (not considering the *HoxD* locus, from which the transgene originated). In the first breakpoint analysis, reads not mapping on their entire length were mapped onto mm10 in local mode (i.e., allowing a segment of each read to not match). All reads containing a NlaIII site (CATG) were filtered out to exclude digestion-ligation events as relevant hybrid junctions, and the coverage was computed from the remaining reads (called CATG-filtered unmapped reads) (see output in Fig 2C, bottom tracks). In a second, more read-conservative breakpoint analysis, reads not mapping on their entire length onto mm10 were retrieved and were split at NlaIII sites. All split reads of more than 25 bp not mapping on their entire length to both mm10 and the transgene vector (called CATG-split unmapped reads) were mapped on mm10 and the transgene vector in local mode, followed by coverage computation (see output in S3A Fig). At last, and for both analyses, the resulting reads were inspected going from the mapped part (known) to the unmapped part (unknown) in order to determine which sequences were brought together through the computation of an average hybrid sequence from all reads displaying a particular connection (see examples in S3C Fig).

### qPCR transgene quantification

Individual ear punches from adult mice were digested in proteinase K for 48 hours, followed by heat-inactivation of the enzyme at 96˚C. gDNA was purified using phenol-chloroform extraction and ethanol precipitation. The qPCR was performed using PowerUp SYBR Green Master Mix (Thermo Fisher Scientific, A25742) in a QuantStudio 5 Real-Time PCR device (Thermo Fisher Scientific). The primers used are listed in S2 Table. For each sample, the results were normalized to the value of *Aldh1a2* using the ΔCt method and outliers were discarded. The qPCR quantification shown in S2A Fig was produced using GraphPad Prism 8 and represents the values relative to the wild-type ($2^{-\Delta\Delta Ct}$) multiplied by two in order to reflect absolute allele counts for each qPCR target region.

### Control-FREEC transgene quantification

Copy number quantification was performed using Control-FREEC version 11.5 [36]. The signal from the ectopic CS38-40 (test dataset) was computed based on the *TgN(38–40)* total input gDNA data of the ChIPmentation experiment (see ChIPmentation below). Next, the signal from the endogenous region CS38-40 (control dataset) was created by pooling total input gDNA data of four samples that were all TgN(38–40)-negative and wild-type for region CS38-40 in chromosome 2, each processed independently. For both test and control datasets, the Control-FREEC signal, expressed as the number of reads scored for non-overlapping genomic windows of given sizes [49], was calculated along a 7 Mb region of chromosome 2 including the *HoxD* locus (chr2:71000000–78000000). The above analysis was carried out using two different window sizes: 1 and 2 kb (see S2B Fig). Then, the software calculated the test/control signal ratio, that is the number of reads from test divided by the number of reads from control, for each window, multiplied by two in order to obtain absolute allele counts. At last, the Control-FREEC software evaluated copy numbers along the 7 Mb chr2 region starting from the (test/control)·2 ratio by a maximum (log-)likelihood estimation [36].

### MinION-nCATS

MinION-nCATS was performed as in [37], following the Cas-mediated PCR-free enrichment protocol (Oxford Nanopore Technologies) with a tiling approach. Single-guide RNAs (sgRNAs) were used instead of cr:tracrRNAs for target enrichment. Multiple pairs of sgRNAs were designed on the target region (Fig 2D, bottom) with Benchling (https://www.benchling.

com/) and were converted into EnGen-compatible DNA oligos (listed in S3 Table) using
NEBioCalculator (http://nebiocalculator.neb.com/#!/sgrna). sgRNAs were produced using the
EnGen sgRNA Synthesis Kit, *S. pyogenes* (NEB, E3322) following the manufacturer's instruc-
tions. Two distinct pools of sgRNAs and Alt-R *S. pyogenes* HiFi Cas9 nuclease V3 (IDT) were
assembled into Cas9 ribonucleoprotein complexes. High molecular weight genomic DNA
(HMW gDNA) was prepared as described hereafter, starting from a single E13.5 *TgN(38–40)/
TgN(38–40); del(CS38-40)*⁻/⁻ headless and tailless embryo. The sample was proteinase K
digested in digestion buffer (50 mM Tris-HCl pH 8, 10 mM EDTA pH 8, 200 mM NaCl, 0.5%
SDS) at 55˚C while shaking for 48 hours. HMW gDNA was purified with two successive
rounds of phenol-chloroform extraction, followed by ethanol precipitation. Size selection was
carried out using 0.8x SPRI magnetic beads. nCATS was performed in two independent Cas9-
mediated release reactions corresponding to the two different pools of sgRNAs and the prod-
ucts from both reactions were pooled together prior to sequencing on MIN106D flow cell.
MinION output data in fast5 format were converted to fastq using the Guppy basecaller (ver-
sion 3.1.5) (Oxford Nanopore Technologies) and the reads were mapped with minimap2 (ver-
sion 2.15) onto mm10 and the TLA-derived configuration that assumed a 63.2 kb integration,
comprising one entire copy of region CS38-40 followed by the partial CS38 segment, after
position chr10:97019221 (mm10) (see Fig 2D, top). Reads mapping on the integration site
(mm10, chr10:97018026–97020425) or transgene region (chr2:75122684–75160161 of mm10)
sequence components of the *TgN(38–40)* mutant construction were converted from fastq to
fasta in order to produce the dot plots displayed in Fig 2E. This was achieved using a Perl script
as in [50], with the following modification: 20 bp of the MinION reads were tested against the
reference for 5 bp-sliding windows and only 20-mers completely identical to unique 20-mers
in the reference were kept. The output was then processed in R (www.r-project.org).

## ChIPmentation

ChIPmentation was performed using the protocol of [51]. Tissues were crosslinked for 15 min-
utes and processed as in [52]. Four *TgN(38–40)/Wt; del(CS38-40)*⁻/⁻ E12.5 whole limbs were
used for the rest of the procedure. The description of experimental replicates is given in S8
Table. A fraction of the samples (see total input DNA below) was preserved to evaluate the effi-
ciency of the chromatin immunoprecipitation by qPCR prior to sequencing and for the Con-
trol-FREEC analysis. Antibodies (CTCF, Active Motif 61311 or RAD21, Abcam ab992) were
incubated with Dynabeads Protein A (Thermo Fisher Scientific, 10001D) for 3 hours on a
rotating wheel at 4˚C. Chromatin immunoprecipitation was performed overnight, followed by
tagmentation for 2 minutes at 37˚C. DNA libraries were sequenced as 50 bp single-end reads
with an Illumina HiSeq 4000. ChIPmentation data analysis was performed as described in
[52], mapping the reads on either the mm10 reference genome or *TgN(38–40)* custom
genome, with the following adaptation: all reads were kept for the *TgN(38–40)* mapping and
hence those aligning to the duplicated sequences were not discarded. These reads were ran-
domly attributed to one or the other location. Due to different signal-to-noise ratios of wild-
type and mutant data, the total enrichment of the CTCF ChIP signal was different on the nor-
malized coverage tracks. To help in the visual comparison of the different CTCF ChIP tracks,
we added a horizontal dashed line at values 3.5 and 1 for the wild-type and mutant ChIP tracks,
which correspond to the height of their respective CS40 CTCF peaks at chromosome 2 (see Fig
3A). We kept the same ratios and placed the dashed lines at 8 and 2.3 in the other loci (see S4
Fig) when comparing the height of other CTCF peaks in the genome. Peak calling of CTCF and
RAD21 was achieved using the MACS2 algorithm (Galaxy Version 2.1.1.20160309.3 with
default parameters) [53,54] on each replicate and the union of the peaks obtained in each

replicate was used in the figures. CTCF site orientation was determined using CTCFBSDB 2.0 (http://insulatordb.uthsc.edu/) [55] with MIT_LM7 motif position weight matrix.

## 4C-seq

4C-seq was performed as in [56]. For each genotype (i.e. wild-type or homozygous *TgN(38–40)*), samples corresponding to twelve E12.5 whole limbs were used as starting material. Inverse PCR was performed using 100 ng template DNA (for 14 reactions in total) and the viewpoint-directed primers listed in S2 Table. Libraries were multiplexed and sequenced to obtain 100 bp single-end reads with an Illumina HiSeq 2500. 4C-seq data analysis was performed as described on the HTSstation web interface (http://htsstation.epfl.ch) [57]. For both CS38 and CS40, the viewpoint was defined as the entire segment located between the two copies of CS38 (i.e., chr10:97036373–97082540, custom genome *TgN(38–40)*). The resulting scores were normalized to the mean score of fragments mapping within 10 Mb around the viewpoint and the signal was smoothened per 11 fragments. All 4C-seq mapped reads and fragment distribution are summarized in S6 and S7 Tables.

## Hi-C

Hi-C was performed as in [5] and [58]. For each genotype (i.e. wild-type or *TgN(38–40)/TgN(38–40); del(CS38-40)$^{-/-}$*), one sample corresponding to four E12.5 whole limbs was used as starting material. Hi-C libraries were multiplexed and sequenced so as to obtain 75/75 bp paired-end reads with an Illumina NextSeq (first run, 80 million reads per sample) or HiSeq 4000 (second run, idem). Hi-C data analysis was as in [58], with some modifications. HiCUP (version 0.7.3) [59] was applied providing either the mm10 reference genome, or the custom genome of *TgN(38–40)*. For the *TgN(38–40)* mapping, the pipeline was adapted in order to prevent removal of reads aligning to the duplicated sequences of this line. All valid hybrid pairs were kept since no MAPQ filter was applied. Analysis of all hybrid pairs resulted in a Hi-C matrix binned at 40 kb, which was further processed using cooler (version 0.8.10) [60] for balancing normalization. All Hi-C sequencing outputs are summarized in S5 Table. TAD or boundary identification in Figs 1, 3 and 4 and S1 Table was done using the hicFindTADs tool from HiCExplorer suite (version 3.6) [61–63] with a fixed window size of either 240 kb, 320 kb, 480 kb or 800 kb and applying Bonferroni p-value correction. To enable direct comparison of the Hi-C maps in Fig 4, the wild-type data were mapped on the mutant genome. However, as the effect of ca. 60 kb extra distance could thereby potentially be underestimated, we adapted the wild-type contact values according to the value of alpha (-0.42) obtained in S5 Fig, so as to faithfully reproduce the distance effect in the DNA stretch beyond the insertion site.

## WISH

Whole-mount *in situ* hybridization (WISH) was performed as in [64]. The *Btg1* RNA probe was generated by cloning cDNA of retrotranscribed RNA obtained from E9.5 whole embryos and using the following primer pair (forward: CTTTGGGTGGGCTCCTCT; reverse: TGGTGGTTTGTGGGAAAGA). To allow for direct comparison, all WISH experiments were done with E12.5 embryos of comparable sizes and were treated together in the same tubes. Pictures were taken with a Leica M205 FCA microscope equipped with a DFC 7000 T camera and were processed with Adobe Photoshop.

## RT-qPCR

Distal forelimbs and livers were dissected from E12.5 wild-type (n = 31) and hemizygous embryos (*TgN(38–40)/Wt*) (n = 27) and kept in RNAlater (invitrogen) at -80˚C. RNA was

extracted with the RNAeasy Micro kit (QIAGEN) after shredding the tissue by pipetting it through a syringe with 27G needle. Reverse transcription was carried out using the GoScript Reverse Transcription System (Promega) and cDNA was amplified cyclically in a QuantStudio 5 Real-Time PCR device (Thermo Fisher Scientific). mRNA levels were referenced to *Actb*. Plotting and statistical analysis (Welch's t-test) were performed with Prism software. The primers used for these experiments are listed in S2 Table. All raw Ct values are listed as S10 Table.

### Sequencing data analysis and display

Chromatin immunoprecipitation (ChIP-seq and ChIPmentation) data were analyzed on our Galaxy platform [65]. Chromosome conformation capture (4C-seq and Hi-C), MinION and TLA data, as well as total input DNA data for Control-FREEC were analyzed through the Scientific IT and Application Support Center of the Ecole Polytechnique Fédérale de Lausanne (EPFL). Data were plotted using the pyGenomeTracks visualization tool (https://github.com/deeptools/pyGenomeTracks) [61,66]. Gene annotations shown in Figs 1, 3 and 4 were retrieved from GENCODE (GRCm38-mm10, VM23 protein-coding). The scripts used to generate all TLA, MinION, 4C-seq and Hi-C outputs as well as those used to produce NGS data figures were deposited in GitHub (https://github.com/lldelisle/scriptsForWilleminEtAl2021). All figures were processed with Adobe Illustrator.

### *TgN(38–40) in silico* mutant genome reconstruction

As depicted in S3B Fig, the final *in silico TgN(38–40)* mutant genome reconstruction was built by inserting a 63812 bp sequence comprising (1) the entire TgN(38–40) fosmid (vector and chr2:75122684–75160161 of mm10), (2) an additional fragment of TgN(38–40) extending towards the CTCF of CS38 (see S3C Fig for details) and (3) the 602 bp duplicated region (mm10, chr10:97019222–97019824) inside chromosome 10 at position chr10:97019824 using the SeqinR package [67]. The sequence of the *TgN(38–40)* mutant chromosome 10 is available at (https://zenodo.org/record/4292337). The *TgN(38–40)* mutant genome was completed by adding wild-type chromosomes of mm10 (retrieved from UCSC), as well as the *del(CS38-40)* mutant chromosome 2 available at (https://zenodo.org/record/3826913) [18].

### Quantifications of 4C-seq and Hi-C contact changes over the host TAD

The quantifications of 4C-seq contacts shown in Fig 3D were performed by summing non-smoothed 4C-seq scores, mapped onto the custom genome of *TgN(38–40)*, over either the left (chr10:96120001–97019221, custom genome *TgN(38–40)*) or right segment (chr10:97083637–97400000, custom genome *TgN(38–40)*) of the host TAD. Both the integration and the 602 bp duplication of chromosome 10 were excluded from the analysis. The resulting values were normalized by the one obtained for the entire host TAD, for each genotype, and the fold change (fc) was computed as follows: fc = (*TgN(38–40)-Wt*)/*Wt*. This analysis was performed in R (https://www.r-project.org/). The quantification of Hi-C contacts in Fig 4D was achieved by retrieving the value of each bin in the new inter-sub-TAD space (chr10:96120000–97000000 and chr10:97080000–97400000, see dashed box) and the fc was computed as follows: fc = (mean(*TgN(38–40)*)-mean(*Wt*))/mean(*Wt*). The p-value was obtained by a Mann-Whitney U-test. This analysis was performed in Python. All quantification scripts are available in the GitHub repository (https://github.com/lldelisle/scriptsForWilleminEtAl2021).

## Supporting information

**S1 Fig. Initial characterization of the *TgN(38–40)* mutant configuration by TLA (related to Fig 2).** (A) TLA signal mapped over all mm10 chromosomes (y-axis data range: 0–5000). Green arrowhead indicates region CS38-40, which composes the transgene, in chromosome 2. Brown arrowhead highlights the integration site in chromosome 10. Asterisk shows an artefactual peak of signal matching a satellite repeat. (B) TLA signal over the fosmid sequence. Below, fosmid scheme. (C) Assessment of all three different possible tandem configurations: tail-to-head, tail-to-tail and head-to-head. Coverage (top) and individual reads (bottom) supporting or dismissing each configuration. Candidate tandem connections are highlighted by a red dotted line. All data displayed in this figure were obtained using the vector viewpoint and correspond to the end-to-end coverage.
(EPS)

**S2 Fig. Transgene quantifications using qPCR and Control-FREEC (related to Fig 2).** (A) qPCR of samples that were either wild-type (blue circles), heterozygous for the deletion of the endogenous region CS38-40 in chromosome 2 (*del(CS38-40)*[+/-], green squares), or hemizygous for the integration (*TgN(38–40)/Wt*, red triangles). Transgene targets: CS38, CS39 and CS40a; control target: *Hoxd8d9*. Vertical axis reflects absolute allele counts. Means are indicated by solid black bars and values are shown above. (B) Control-FREEC transgene quantification using non-overlapping windows (w) of size 1 or 2 kb. Both (test/control)·2 signal ratio and copy numbers estimations (copy #) are shown along region CS38-40 (mm10 coordinates). The copy # signal represents absolute allele counts and the values are indicated in white within the corresponding tracks. Bottom, extension of the TgN(38–40) construct and *del(CS38-40)* background. In both panels, red and yellow lines indicate expected values for one and a half or two and a half fosmid copies, respectively.
(EPS)

**S3 Fig. Base-pair map of the *TgN(38–40)* mutant genome (related to Fig 2).** (A) TLA reanalysis revealing left and right integration breakpoints (red and green arrows, respectively). The region displayed (mm10, chr10:97019018–97020046) is centered around the validated integration site (mm10, chr10:97019222–97019824). Both end-to-end and CATG-split unmapped coverages are shown. TLA restriction sites are shown at the bottom of the panel. (B) Schematic reconstruction of the TgN(38–40) integration. (C) Connections between chromosome 10 and the construct (left breakpoint, red; right breakpoint, green). Sequences are color-coded and/or underlined according to their origin. All identified connections were found in such a sequence configuration that the Watson strands of the transgene (tg) and chromosome 10 were fused. The asterisk indicates the limits of the 602 bp duplication.
(EPS)

**S4 Fig.** ChIP of CTCF in several control loci at (A) chromosome 1, (B) chromosome 8 and (C) chromosome 19 (related to Fig 3). Dashed horizontal lines are displayed for indirect comparison with CTCF binding at region CS38-40 (see Materials and Methods).
(EPS)

**S5 Fig. Frequency of interactions and genomic distances (related to Fig 4).** (A) Logarithmic relationship between interaction probability and increasing genomic distances computed from the Hi-C data of wild-type limbs. These calculations were applied genome-wide, to an aggregate of all TADs in the genome and to individual TADs including the *Btg1* TAD at chromosome 10 and six other TADs in the same chromosome. Values of alpha and R-value are shown

for each condition. (B) Hi-C map depicting TADs 1–8 analyzed in (A).
(EPS)

**S6 Fig. Global *Btg1* expression changes upon reorganization of the host chromatin land-scape (related to Figs 3 and 4).** (A) *Btg1* WISH in wild-type and *TgN(38–40)* mutant embryos at E9.5. Heads were partially severed and used for genotyping. Arrowheads point to the area where the presumptive limb bud is located. Scale bar: 500 μm. (B) *Btg1* WISH in wild-type and *TgN(38–40)* homozygous embryos at E12.5. LPM: Lateral plate mesoderm. FM/WP: Facial mesenchyme and whisker pads. Scale bar: 1 mm. (C) Magnified pictures of mammary buds for the same embryos as in panel B. The position of the forelimbs, which were removed for easier mammary bud visualization, is highlighted by a dotted line. Scale bar: 300 μm. The proportion of embryos displaying equivalent patterns in each experiment is shown. Empty arrowheads indicate changes in expression compared to bold arrowheads.
(TIF)

**S1 Table. Identification of topological boundaries using various window sizes.** Source data (for different genotypes and mapped genomes) and TAD calling window sizes (w) are indicated. At the *HoxD* locus, *Atf2* is the left boundary of the C-DOM and *Hnrnpa3* is the right boundary of the T-DOM. Left_Bd and Right_Bd are respectively the left and right boundaries of the TAD hosting the TgN(38–40) construct in chromosome 10 (see Figs 3B and 4A).
(DOCX)

**S2 Table. List of TLA, qPCR, 4C-seq and RT-qPCR primers used in this study.** For the 4C-seq primers, Illumina Solexa sequencing adapters are indicated in red (long adapter) or blue (short adapter). For both CS38 and CS40 viewpoints, a 4 bp barcode (underlined) was present between the long sequencing adapter and the rest of the primer. F: forward. R: reverse. iF: inverse forward. iR: inverse reverse.
(DOCX)

**S3 Table. EnGen-compatible DNA oligos used as templates for sgRNA production.** The name of the oligos indicates their approximate position on the *TgN(38–40)* transgene or sur-rounding regions: upstream of the transgene in chr 10 (up); one quarter (1/4), halfway (1/2) or three quarters (3/4) into the transgene; 3' part of transgene vector (pEpi3) and downstream of the transgene in chr10 (down). Two pairs of primers (1 or 2) were design to target the same region. Underlined, sequence matching target DNA. Red, sequence of the T7 promoter for sgRNA production. Cyan, RNA scaffold for the Cas9 enzyme. A G was added (highlighted in black) when not present in the original target sequence, to ensure efficient sgRNA transcription.
(DOCX)

**S4 Table. Summary of the MinION sequencing output.** Target reads are reads mapping to the construct or the integration site. Construct coordinates taken into consideration: chr2:75123000–75160000 (mm10). Considered integration site: chr10:97018700–97019300 (mm10). *Ratio of bases of the mutant construction (64,420 bp; see Fig 2D) relative to the hap-loid mouse genome (around 2.6 Gb): 2.457384e-05.
(DOCX)

**S5 Table. Summary of the Hi-C sequencing output.** Two samples were sequenced: TgN3840 (mutant) and Wt (control). Both were subsequently mapped either on mm10 wild-type mouse genome or the custom TgN(38–40) genome. Total reads correspond to all raw reads obtained from the sequencing platform. Cis-far reads correspond to intra-chromosomal interactions

located further than 10 kb. All sequencing outputs are shown as base pairs (bp).
(DOCX)

**S6 Table. Summary of mapped reads from 4C-seq experiments.**
(DOCX)

**S7 Table. Summary of 4C-seq fragment distribution.**
(DOCX)

**S8 Table. Biological replicates of the ChIP-seq and ChIPmentation (ChIPm) experiments.**
Wild-type ChIP-seq data of CTCF, RAD21, and H3K27ac were retrieved from a previous publication of our group (see Data availability). WL: whole limbs. DFL: distal forelimbs.
(DOCX)

**S9 Table. Genotypes of 4C-seq and Hi-C samples.** Genotypes of 4C-seq samples are colored in the same way than the corresponding tracks of Fig 3. WL: whole limbs (including both forelimbs and hindlimbs).
(DOCX)

**S10 Table. Spreadsheets of Ct values coming from the RT-qPCR experiments done in limbs and livers from wild-type and hemizygous embryos (*TgN(38–40)/Wt*).**
(XLSX)

## Acknowledgments

We would like to thank Thi Hanh Nguyen Huynh, Sandra Gitto and Bénédicte Mascrez for help with mice breeding and genotyping; Mylène Docquier, Brice Petit, Didier Chollet and Christelle Barraclough from the Geneva Genomics Platform, as well as Bastien Mangeat, Elisa Cora and Lionel Sylvain Ponsonnet from the EPFL Gene Expression Core Facility. We are grateful to all members of the Duboule laboratories for comments and discussions.

## Author Contributions

**Conceptualization:** Denis Duboule, Eddie Rodriguez-Carballo.

**Data curation:** Lucille Lopez-Delisle.

**Formal analysis:** Andréa Willemin, Lucille Lopez-Delisle, Eddie Rodriguez-Carballo.

**Funding acquisition:** Denis Duboule.

**Investigation:** Andréa Willemin, Christopher Chase Bolt, Marie-Laure Gadolini, Eddie Rodriguez-Carballo.

**Methodology:** Andréa Willemin, Christopher Chase Bolt, Eddie Rodriguez-Carballo.

**Project administration:** Denis Duboule, Eddie Rodriguez-Carballo.

**Resources:** Marie-Laure Gadolini.

**Software:** Lucille Lopez-Delisle.

**Supervision:** Denis Duboule, Eddie Rodriguez-Carballo.

**Validation:** Andréa Willemin.

**Writing – original draft:** Andréa Willemin, Lucille Lopez-Delisle, Denis Duboule, Eddie Rodriguez-Carballo.

**Writing – review & editing:** Christopher Chase Bolt.

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
