## [Decision Letter · Decision Letter 0]

17 Mar 2021

Dear Dr Duboule,

Thank you very much for submitting your Research Article entitled 'CONTEXT-INDEPENDENT FUNCTION OF A CHROMATIN BOUNDARY IN VIVO' to PLOS Genetics.

The manuscript was fully evaluated at the editorial level and by independent peer reviewers. The reviewers appreciated the attention to an important problem, but raised some substantial concerns about the current manuscript. Based on the reviews, we will not be able to accept this version of the manuscript, but we would be willing to review a revised version. We cannot, of course, promise publication at that time.

If you decide to revise the manuscript for further consideration at PLOS Genetics, please aim to resubmit within the next 60 days, unless it will take extra time to address the concerns of the reviewers, in which case we would appreciate an expected resubmission date by email to plosgenetics@plos.org.

[LINK]

We are sorry that we cannot be more positive about your manuscript at this stage. Please do not hesitate to contact us if you have any concerns or questions.

Yours sincerely,

Stefan Mundlos

Associate Editor

PLOS Genetics

Gregory Barsh

Editor-in-Chief

PLOS Genetics

Reviewer's Responses to Questions

**Comments to the Authors:**

Reviewer #1: This is an interesting study reporting the first accurate characterization of the in vivo structural and functional effects of an ectopic insertion of genomic sequence containing a TAD boundary. The experiments meet high technical standards and in addition to providing interesting insight into the role of chromosome topology, also highlight the fact that every ectopic insertion deserves very careful examination before its structural and functional impact can be assessed.

The main limitation of the manuscript is that only one ectopic location and one version of the ectopic sequence are analyzed. This makes it hard to infer how generally the results can be extrapolated, and also to formally conclude on the mechanistic role of CTCF sites and loop extrusion in establishing the interactions observed at the ectopic location. I do not think however that this should be held up against the manuscript, given the complexity and timescales of in vivo experiments.

I only have a very small number of comments and suggestions (#1 is crucial though):

1) I would like to point the authors to the fact that the differential analysis of Hi-C data (Fig. 4) is not formally correct. Simply inserting a blank space in the WT map does not account for the correct decay of contact probabilities that would be generated by ~60kb of extra genomic sequence, even in the complete absence of CTCF sites or any other sequences that can create specific interactions. This is due to the power-law behavior of contact probabilities and 60 kb can make a big difference!

In order to conclude that sequence identity (rather than length) is responsible for the appearance of an extra sub-TAD boundary, the transgenic sample should be compared to a better (although not perfect) control where the WT counts across the insertion are rescaled by the average power-law decay inside the original TAD, which can be extracted from the WT Hi-C map. The formally correct control, which however seems out of experimental reach in the context of this study, would be a line carrying the homozygous insertion of the same ectopic sequence without CTCF sites.

2) In Fig. 4C it is unclear what the scalebar represents.

3) It would be very interesting to explore to which extent the amount of Btg1 downregulation depends on the amount of physical insulation provided by the ectopic boundary. For example does it depend on the number of CTCF sites in the boundary, as recently reported (Huang et al., https://www.biorxiv.org/content/10.1101/2020.07.07.192526v1)? The homozygous TgN(38-40) line provides an ideal background for generating CRISPR deletions affecting the number of CTCF sites in the transgenic boundary.

4) It would be nice to see higher magnification Hi-C maps – it looks like the number of reads in the experiments is not reported (should be included in a revised version) but from a qualitative assessment of the maps provided, this could be feasible.

Reviewer #2: In this work the authors use mouse transgenesis to randomly insert a previously characterized TAD subdomain boundary from the HoxD locus randomly into the mouse genome. First, the authors in detail characterize the transgene insertion in the Btg1 locus using a elaborate combination of state of the art approaches and find that the boundary to be inserted in a partial tandem copy within the Btg1 TAD. Following this, the authors characterize the effects of the inserted boundary on 3D chromatin structure and find that also at the new location the boundary element consisting of 4 CTCF sites (3+1 duplicated) serves as a sub-domain boundary (and not a "full/strong" boundary) that allows some contacts beyond the insertion point. Using in situ hybridizations of E12.5 embryos the authors show that the insertions leads to an overall reduced expression level of Btg1 without major changes in expression pattern.

The in-depth analysis of the transgene insertion using multiple analogous methods can serve as a case example of how to comprehensively characterize new transgenes, especially for studies that "assume" that the knock-in happened as predicted in theory.

The characterization of the 3D chromatin effects that originate from the sub-domain boundary are interesting, although limited in scope, given that only one alternative boundary position is analyzed. Stik et al in HAP1 cells and Huang et al (https://www.biorxiv.org/content/10.1101/2020.07.07.192526v1) in mouse embryonic stem cells have taken an alternative approach and characterized several integrations in cell culture systems. However, the in vivo approach taken here enables a more comprehensive functional readout at an organismal level similar to our own boundary repositioning at the Sox9 locus.

Therefore, the characterization of the gene regulatory effects of the boundary insertion should be more comprehensive in order to fully realize the potential of in vivo approach that the authors undertook. I would suggest including additional data that should be available or easily attainable to the authors (if the mouse line is still available) prior to publication.

Major comments:

Regulatory effects 1:

The authors should expand the regulatory consequences of the CS38-40 insertion into the Btg1 domain on Btg1 expression. As the authors have previously shown (Andrey et al 2013, Rodriguez-Carballo et al 2020), CS39 comprises a potent limb bud enhancer that is particularly active at stages prior to E12.5 and the TSS of two lncRNAs. If the mouse lines are available, it would be very informative to determine whether this leads to increased Btg1 levels at these early stages, assessed through WISH, but preferably also through more quantitative methods. Also, do lncRNA transcripts arise from the knock-in construct?

Regulatory effects 2:

Using publicly available data, such as chromatin ChIP-seq from the mouse ENCODE dataset, the authors could identify embryonic or adult tissues where the majority of regulatory elements is not cut off from Btg1 by the boundary insertion (kidney? heart? brain?). In these tissues (that could be obtained from the same crosses as used above) the loss of Btg1 expression should be less that in E12.5 limb buds. This data would strengthen the observation that the inserted boundary cuts off Btg1 from its major limb enhancers and the right Btg1 TAD boundary.

Reviewer #3: Defining the characteristics that are required for the formation of topological boundaries has received significant attention, including genetic experiments, biochemical analyses, computational, and modeling studies. Analysis of TAD boundaries has largely focused on use of deletions of chromosomal regions, or acute deletion of specific motifs such as CTCF binding sites, but relatively few studies have examined if the underlying DNA sequence can encode a boundary in different chromatin contexts.

In this manuscript the authors ask whether repositioning of a strong, CTCF-bound element from the HoxD cluster retains its potential to insulate chromatin when placed in an ectopic genomic context. The strength of this manuscript is in the novelty of performing this experiment using an in vivo context. However, as the boundary is moved to only one ectopic context, it cannot be determined if the results discussed herein can be extrapolated to additional regulatory sequences, for example boundaries with weaker insulation scores, or to different chromatin contexts.

Overall, the authors included excellent characterization of the integration site and the type of insertion event and have shown that the Tg mice of the large fragment acts similarly at the insertion site to its role in the endogenous environment. I think this work is sufficiently novel and would be of interest to the community, provided that the following comments are addressed:

Major points:

1. Title

I feel that the claim in the title is a bit misleading – there is only one integration site, so it is a bit of a stretch to conclude that the context doesn’t matter. In addition, the integrated sequence contains 4 CTCF reverse sites – would it still act as a boundary if it was in a context with no adjacent CTCF sites (for example in a compartmental domain or in another species where loop-extrusion loops are not so prominent such as drosophila)? To be clear, I don’t think that these experiments are required, I would just suggest to make the title more precise to avoid any misleading claims.

2. Quantification of changes in Btg1 expression

Changes in Btg1 expression levels are difficult to quantify using WISH, it would be better to use qPCR as orthogonal approach, which could be used to examine the reduction in the context of both the homozygote and heterozygous animals.

3. Mechanism of Btg1 repression

One alternative to the hypothesis that Btg1 expression is downregulated due to the topological isolation of a putative enhancer is proximity spreading of a repressive mark such as H3K27me3. Indeed, the authors have previously shown that this region insertion is heavily decorated in Polycomb marks in its native context (Rodriguez-Carballo et al., 2019 – Figure 3). I think that this is an important conceptual distinction (linear spreading vs disruption of 3D landscape) and resolving it may enhance the conceptual novelty of the paper. H3K27me3 ChIP-seq in the Tg vs WT would resolve this question.

Minor Points:

- Please add mapping and QC stats for the Hi-C/4C experiments to better estimate the resolution

- Figure1: It would be great to also add the insulation score (similar to Figure4) to quantitatively show the loss of insulation upon deletion

- Figure3: Please show the CTCF ChIP signal in an unrelated control region (wt vs TgN(38-40), to ensure that the difference in intensity is indeed due to the presence of the extra CS38 site

- Figure4: it would be great to also show the H3K27ac ChIP-seq signal at the integration locus, in order to visualize the putative enhancer (instead of just mentioning it in the discussion).

- Showing the insulation score in Fig 4 as an overlay between WT and the Tg line would make the comparison much easier for the reader.

**Have all data underlying the figures and results presented in the manuscript been provided?**

Reviewer #1: Yes

Reviewer #2: Yes

Reviewer #3: Yes

PLOS authors have the option to publish the peer review history of their article (what does this mean?). If published, this will include your full peer review and any attached files.

Reviewer #1: No

Reviewer #2: **Yes: **Daniel Ibrahim

Reviewer #3: No

---

## [Decision Letter · Decision Letter 1]

30 Jun 2021

Dear Dr Duboule,

We are pleased to inform you that your manuscript entitled "INDUCTION OF A CHROMATIN BOUNDARY IN VIVO UPON INSERTION OF A TAD BORDER" has been editorially accepted for publication in PLOS Genetics. Congratulations!

Yours sincerely,

Stefan Mundlos

Associate Editor

PLOS Genetics

Gregory Barsh

Editor-in-Chief

PLOS Genetics

Comments from the reviewers (if applicable):

Reviewer's Responses to Questions

**Comments to the Authors:**

Reviewer #2: The authors have adressed all reviewer comments and I support publication of the manuscript.

**Have all data underlying the figures and results presented in the manuscript been provided?**

Reviewer #2: Yes

PLOS authors have the option to publish the peer review history of their article (what does this mean?). If published, this will include your full peer review and any attached files.

Reviewer #2: **Yes: **Daniel Ibrahim

**Data Deposition**

http://datadryad.org/submit?journalID=pgenetics&manu=PGENETICS-D-21-00201R1

**Press Queries**

---

## [Editor Report · Acceptance letter]

14 Jul 2021

PGENETICS-D-21-00201R1 

INDUCTION OF A CHROMATIN BOUNDARY IN VIVO UPON INSERTION OF A TAD BORDER 

Dear Dr Duboule, 

We are pleased to inform you that your manuscript entitled "INDUCTION OF A CHROMATIN BOUNDARY IN VIVO UPON INSERTION OF A TAD BORDER" has been formally accepted for publication in PLOS Genetics! Your manuscript is now with our production department and you will be notified of the publication date in due course.

With kind regards,

Zsofi Zombor

PLOS Genetics

On behalf of:
